# Extensive Analysis on the Effects of Post-Deposition Annealing for ALD-Deposited Al_2_O_3_ on an n-Type Silicon Substrate

**DOI:** 10.3390/ma14123328

**Published:** 2021-06-16

**Authors:** Atish Bhattacharjee, Tae-Woo Kim

**Affiliations:** School of Electrical Engineering, University of Ulsan, Ulsan 44610, Korea; atishbhattacharjee27@gmail.com

**Keywords:** Al_2_O_3_, ALD, constant voltage stress, diffusion, MOSCAP, PDA

## Abstract

In this study, an investigation was performed on the properties of atomic-layer-deposited aluminum oxide (Al_2_O_3_) on an n-type silicon (n-Si) substrate based on the effect of post-deposition heat treatment, which was speckled according to ambient temperature and treatment applied time. Based on these dealings, a series of distinctions for extracted capacitance and dielectric constant, hysteresis was performed on annealed and nonannealed samples. The interface and border trap responses, including stress behavior after an application of constant voltage for a specific time and surface morphology by X-ray diffraction (XRD) technique, were also analyzed between the two above-mentioned sample types. Based on observation, the annealed samples showed superior performance in every aspect compared with the nonannealed ones. Some unusual behaviors after high annealing temperature were found, and the explanation is the ion diffusion from oxide layer towards the semiconductor. Since a constant voltage stress was not widely used on the metal–oxide–semiconductor capacitor (MOSCAP), this analysis was determined to reveal a new dimension of post-deposition annealing condition for the Al/Al_2_O_3_/n-Si gate stack.

## 1. Introduction

A large band gap and an exalted barrier height between the dielectric and Si has essential chemical, along with the thermal steadiness of the conventional dielectric material silicon dioxide (SiO_2_), on silicon (Si) wafers [1,2,3]. However, the applicable SiO_2_ layer thickness (which functions as the insulating layer) reaches its maximum owing to the miniaturization of metal–oxide–semiconductor field-effect transistors (MOSFETs). Because of direct tunneling through the SiO_2_ film, low-power application devices mostly agonize from high leakage currents [4]. Nevertheless, the leakage current and equivalent oxide thickness (EOT), which are associated with the speed of the transistor, can be decreased with a denser oxide layer with a high dielectric constant (k). Among the different deposition techniques, such as physical vapor deposition, chemical vapor deposition (CVD), and atomic layer deposition (ALD) for high-k oxides, ALD [5,6,7,8,9] is considered the most promising technique for studying the microelectronic and nanotechnological characteristics of samples. Being a subtype of CVD, in ALD, at the same time in the deposition chamber, the precursors and oxidants are not only present but also introduced in a chronological and noncongruent way. The continuous purge gas flow eradicates residual precursors and reactant species between the precursor and oxidant inoculations. The combination of the precursor/oxidant pulses and purge gas flow is acknowledged as a half cycle; the film is deposited through this self-saturating half cycle because the reactions stop once all reactive components on the Si surface are devoured. Throughout this process, the reactions are eradicated by themselves [9]. Therefore, deposition occurs in a cyclic order in ALD, whereas it happens on a time basis in CVD [5]. The thickness of the deposited film can be accustomed to the number of deposition cycles. Thus, the self-saturating nature and cyclic deposition deliver amenable, disciplined, uniform, high-quality, condensed, and pinhole-free thin films with thickness errors of less than 1%; the film growth is autonomous of the precursor and oxidant flux [7,10]. The different deposition procedures (including ALD) are comprehensively compared in [9]. Al_2_O_3_, which is one of the studied high-k oxides, is considered to be one of the most auspicious substitutions for SiO_2_ because of its simple ALD manufacturing process [11]. Al_2_O_3_ has a higher bandgap (approximately 8.8 eV) and band orientation analogous to SiO_2_, good thermal steadiness, and concentrated oxygen and ionic carriage, with a reasonable dielectric constant (k = 6–9) [12,13].

Post-metal annealing (PMA) and post-deposition annealing (PDA) are two methods of rapid thermal annealing (RTA) used to observe the features of MOSCAPs. This observation can be conducted at different temperatures for different durations in an ambient gas environment. For the Al_2_O_3_ MOSCAP, numerous research studies have been conducted on PMA characteristics [14,15,16]. In PMA, after the fabrication of the oxide layer and metal layer on the semiconductor, the MOSCAP is annealed at a specific temperature for a definite time in a suitable ambient gas condition. In PDA, the MOSCAP is annealed after the fabrication of the oxide layer and before the deposition of the metal layer. Researchers have discovered that a slowly increasing PMA temperature makes the capacitance value reduced and the leakage current enlarged. These outcomes signpost that PMA with aluminum electrodes is substantially subtle to the annealing condition compared with PDA [17]. Additionally, PMA ought to have higher value of equivalent oxide thickness (EOT) and interface state density as compared with PDA. Besides, there is no frequency dependence in the case of PDA found in the weak inversion region of capacitance–voltage (C–V) curves of PMA-annealed samples [17]. According to [18,19], PDA can affect the capacitance, hysteresis, dielectric constant, and morphological structure, but constant voltage stress, interface, and border trap characterizations are not clearly stated in those articles. The shift in the threshold voltage after constant voltage stress should generally differ from MOSCAP to MOSCAP based on the annealing conditions (e.g., duration, temperature, and ambient gas environment). In this paper, all these attributes, along with the constant voltage stress for the Al/Al_2_O_3_/n-Si MOSCAP regarding PDA, are discussed.

The flat-band voltage, V_FB_ (which is the gate voltage at which the energy bands in the semiconductor substrate remain horizontal with respect to the semiconductor–dielectric interface and at which the surface potential of the semiconductor substrate is zero), is one of the most critical parameters of a metal–oxide–semiconductor (MOS) structure. The importance of this parameter is attributable to the fact that it is a critical feature of the threshold voltage, V_T_, of the MOS transistor and its diverse uses for other characteristics of MOS structures [20]. The shift of the flat-band voltage in the positive direction in the C–V curve is a prominent problem of Al_2_O_3_ films. The positive shift is due to negative charge traps in the oxide layer induced by Al diffusion from the Al_2_O_3_ layer to the Si substrate [21,22]. When the annealing temperature increases, the diffusion rate increases. In addition, because neither optical nor electrical impulses are applied, free charges must be created by the trapping or detrapping defects inside the dielectric; the energy required to trigger this mechanism can only be provided by temperature (i.e., phonons). The bulky shift of the flat-band voltage upsurges the Coulomb scattering between carriers and trap charges. This causes the catastrophic reduction of the conductance of the carrier. Researchers have stated that the variations in the chemical states and atomic structures of films are strongly linked to these phenomena [21,22,23,24]. Because Al^3+^ diffuses into the Si substrate, the density of positive ions (holes) in the semiconductor increases, which attracts more negative charges (electrons). Thus, n-type Si, which usually functions as the donor, starts to function as the acceptor. Moreover, as the atomic radius of Al (143 pm) is larger than that of Si (118 pm) [25], diffusion is accompanied by some scattering events in usual Si atomic structures. These phenomena may change the attributes of samples that have been annealed at high temperature and are subjected to constant voltage stress.

Constant voltage stress measurements are conducted to determine how a MOSCAP device reacts when a gate bias is applied for a specific duration; in these measurements, unusual properties such as strain instead of stress can occur after certain PDA conditions. In this study, some other hallmarks of MOSCAP characterization, such as interface trap density (D*_it_*) and border trap (N_bt_), are extracted. To determine the D*_it_* of dielectric/semiconductor (SiO_2_/Si) interfaces of MOSCAP structures, researchers developed different approaches in the 1960s [26]. High interface trap densities are the reason for unproductive Fermi level response. In addition, they can cause Fermi level pinning, thereby preventing the successful control of charge carriers in the channel and the comprehension of MOSFETs with good subthreshold slopes and high drive currents, although Si possesses high-quality native oxides with good stability, low leakage, and high breakdown fields. To extract the D*_it_*, the conductance method was used in this study, as recommended in [26].

Owing to the time constant difference between both forms of traps and the border trap density estimate from the C–V hysteresis, which exhibits full re-emission of captured charges during the reverse C–V sweep, the traditional interface trap model does not explain the existence of border trap phenomena. Their existence must be proved based on the dispersion of accumulated frequencies [13,27]. Many researchers have investigated border trap reduction caused by annealing; nevertheless, the annealing environment and its effects have not been studied thoroughly [28,29]. A robust method for border trap extraction is presented in [29,30,31,32]; the same strategy was used in this study.

## 2. Methods and Materials

The Al_2_O_3_ thin film was deposited with a thermal ALD system. The deposition temperature was 250 °C, and an n-type Si (100) substrate (Sehyoung Wafer Co. Ltd., Seoul, Korea) with 1–100 Ω cm resistivity and try-methyl aluminum (TMA) as the precursor for Al_2_O_3_ were used. The Chemical Abstracts Service number of TMA is 75-24-1; it has 99.99% purity according to UP Chemical Co., Ltd. (Pyeongtaek, Korea). H_2_O was used as the oxidizing agent, and argon was the carrier and purge gas. The ALD system consisting of an allocation system with four sets of precursor canisters conducted the thermal ALD process (“Atomic Classic”, CN1, Hwaseong, Korea) at a maximal deposition temperature of 450 °C. The TMA precursor was kept at room temperature. The carrier and purge gas flow rates for Al_2_O_3_ deposition were 0.3 L/min. Moreover, the pulse time was 0.1 s, and the purge time was 20 s. The oxidant (H_2_O) pulse and purge times were 0.1 s and 60 s, respectively. The Al_2_O_3_ layer was deposited with 100 cycles. Before deposition, the substrate was processed with a standard precleaning method with acetone, isopropyl alcohol, and deionized water. Subsequently, the substrate was dried in a N_2_ environment for the prevention of watermark formation on the surface. Eight samples were prepared for oxide layer deposition. The deposition started with pre-argon purging, 100 cycles of Al_2_O_3_ deposition as the second step, and post-argon purging as the third step. The thicknesses of the ALD-deposited films were measured with ellipsometry at an incident angle of 70°. Because all eight samples had been simultaneously covered with Al_2_O_3_, their oxide thicknesses were approximately identical. The average calculated growth per cycle (GPC) was 1.14 Å. For the PDA process of the substrates, the Nextron TM rapid thermal processing system RTP-1200 and Atovac flow and pressure controller GMC1200 were used. The RTA time and temperature were as follows: 300 °C–2 min, 300 °C–5 min, 300 °C–10 min, 400 °C–2 min, 400 °C–5 min, 500 °C–2 min, and 500 °C–5 min; the resulting characteristics were compared with those of as-grown substrates that were not treated with PDA. Instead, they were preannealed at 100 °C for 1 min. To prepare MOSCAP devices, an approximately 150 nm thick Al metal layer was deposited with a thermal evaporator (Korean vacuum thermal evaporator system, KVT-438) on the dielectric to create front electrodes with different areas with a shadow mask: 200, 300, and 400 μm^2^. The same metal layers were deposited without masks as back contacts. The electrical characterization (e.g., the C–V and constant voltage stress measurements) and interface and border trap extractions were performed with a probe station (MSTech 5500), semiconductor device analyzer (Keysight B1500A), waveform generator/fast measurement unit (Keysight B1530A), precision LCR meter (Agilent 4284A), and low-leakage switch mainframe (HP E5250AX-Ray diffraction (XRD) was measured by an X-ray diffractometer (Rigaku Ultima 4) where, Cu Kα radiation (40 kV, λ = 1.54 Å) was used for measuring. Various conditions for preparing the MOSCAPs are shown in Table 1.

## 3. Results and Discussion

Figure 1 demonstrates the C–V characteristics of the Al_2_O_3_ MOS capacitors (PDA) annealed at different temperatures for different times at 1 MHz. The colored curves represent the capacitance values of the MOSCAPs. The MOSCAP annealed at 300 °C for 5 min has the highest capacitance, and the 500 °C–5 min sample has the lowest. The as-grown and other samples have similar values except sample 400 °C–5 min; its capacitance value is lower than that of 300 °C–5 min and higher than those of the others. The C–V curves of the samples are fluctuating towards the right side as PDA time and temperature extend. The curves of the as-grown to 400 °C–5 min samples are close; however, those of the samples annealed at 500 °C shift abundantly. When a C–V curve is shifting towards the positive side, that means the threshold voltage, as well as the flat-band voltage (V_FB_), is also increasing.

Figure 2a shows the V_FB_ value of each sample; it gradually increases with increasing PDA time and temperature. The V_FB_ values of the samples were calculated with the inflection point method using a second derivative of the C–V curve. The point of intersection of the second derivative of the C–V curve with the Vg-axis is zero, which is called the inflection point, as well as the value of flat-band voltage [33]. Most researchers have stated that V_FB_ shifts owing to ion diffusion at the annealing temperature [21,22,23,24]; this results in an additional aluminum silicate layer at the interface of the Si substrate and oxide layer, as presented by the X-ray diffraction (XRD) data in Figure 2b (each sample exhibits peaks at 82.5°) [34]. For the XRD measurements, the annealing time was set to 5 min, and the temperatures were 300, 400, and 500 °C. We measured 2θ from 30° to 90°, but significant difference in the peak analysis, in terms of annealing, was not found except in 2θ = 82.5°. From the literature [34,35,36], we found that this value of 2θ indicates the presence of aluminum silicate. That is why we narrowed down the whole XRD waveform into around 80 degree. The peak variations among the four samples indicate that the increasing temperature promotes diffusion in the interfacial layer. The peak of the sample annealed at 500 °C is much higher than those of the others; this confirms greater diffusion at 500 °C. In the n-Si samples, the shift of V_FB_ to the right means that Al^3+^ ions from the Al_2_O_3_ layer diffuse toward the Si substrate.

Figure 3a compares the dielectric constants of the samples. The 300 °C–5 min sample has the highest dielectric constant because the constant depends on the capacitance value. The dielectric constant is calculated as follows [26]:(1)$$\varepsilon_{high-k}=\frac{C_{\max}\times T_{ox}}{\varepsilon_{0}}$$

Figure 3b presents the hysteresis results. Accordingly, the 300 °C–5 min sample has the lowest hysteresis. In general, the Al_2_O_3_ hysteresis is inversely proportional to the PDA time and temperature. However, above certain temperature and time thresholds, the hysteresis increases; this increase is still lower than that of the as-grown sample (which has no PDA-annealed MOSCAP). The highest capacitance and hysteresis were measured in several devices in each of the MOSCAPs to be sure about the trend. All test and average calculation results and the calculated standard deviation of the seven samples are listed in Table 2 and Table 3. The standard deviation indicates how disperse the data are in relation to the average value. For example, in Table 2, the standard deviation values indicate that the 300 °C–5 min sample has the lowest scattering in terms of hysteresis.

Figure 4a exemplifies the interface trap density (D*_it_*) of the samples and as-grown sample under different treatment conditions; the trap density was calculated with the conductance method by considering the series resistance correction with the following equation:(2)$$Dit=\frac{2.5}{\mathrm{Aq}}\times {\left(\frac{G_{p}}{\omega }\right)}_{\max}$$
where A is the area of the measured device, q the electron charge, G_p_ the parallel conductance, and ω the angular frequency. [26] The D*_it_* values of the as-grown, 300 °C–2 min, 300 °C–5 min, 300 °C–10 min, 400 °C–2 min, 400 °C–5 min, 500 °C–2 min, and 500 °C–5 min samples are 5.8 × 10^11^, 5.6 × 10^11^, 4.94 × 10^11^, 1.32 × 10^11^, 5.06 × 10^11^, 5.1 × 10^11^, 3.7 × 10^11^, and 2.67 × 10^11^ cm^−2^ eV^−1^ because all the samples had identical interfaces and Al_2_O_3_ layer thicknesses and the same pretreatment. The D*_it_* pattern exhibits two characteristics. First, the increasing annealing time decreases D*_it_* (the 5 min time frame presents greater reduction in the density of interface traps than the 2 min case). Second, the interface trap density decreases more at 500 °C than at 300 °C, although there is a little discrepancy in the 400 °C cases. Thus, a high annealing temperature and long annealing time effectively reduce D*_it_*.

Figure 4b presents the extracted border trap densities, N_bt_, of the MOSCAPs. They were determined with the distributed border trap model proposed by Yuan et al. by constructing the best fit between the measured capacitance at the precise voltage in the accumulation region and the capacitance calculated from the model [37]. The overall oxide thickness is segmented into a limited number of quantities in this model. Each quantity reflects a certain oxide capacitance that is proportional to the border trap extent and arranged in series with the semiconductor capacitance in a parallel admittance model. In [30,31], the N_bt_ model and extraction method were thoroughly described. In the extraction process, the effective electron mass of the Al_2_O_3_ film was considered to be 0.23 m_0_, where m_0_ represents the electron rest mass [27,38]; the trap capture/emission time constant τ0 was a fitting parameter. In addition, a one-dimensional Poisson–Schrodinger solver simulation tool (nextnano) was used to calculate the semiconductor capacitance C_s_ at the border trap extraction voltage [39]. Here, almost the same pattern was observed as interface trap density (D*_it_*) in the extracted densities of the border trap. Many of these special oxide traps disappear at high annealing temperature and with longer annealing time, which is supposed to more stoichiometrically change at these conditions. According to Figure 4b, the 300 °C–10 min MOSCAP has the lowest amount of border traps.

Figure 5 shows the V_T_ shifts in the constant voltage stress measurement. The CVS was measured at 1.5 and 2.0 V stress biases for 0, 10, 30, 60, 200, 300, 400, 1000, and 2000 s. The V_T_ shift indicates how much degradation occurred due to stress in the sample. Figure 5a,b indicates that the 300 °C–5 min annealed sample has the lowest dispersion after stress for both stress bias voltages, while the as-grown sample has the highest dispersion, and other MOSCAPs are in the intermediate range. Evidently, the post-deposition heat treatment improves the film quality. However, at a relatively high temperature, the scenario has dramatically changed. According to the insets in Figure 5a, b, in the results of the samples 500 °C–2 min and 500 °C–5 min, the V_T_ shifts left after the stress treatments. This is probably due to excessive ion diffusion from the oxide layer to the n-Si substrate. Because positive Al^3+^ ions diffuse to the Si layer, the electron concentration of the n-doped Si layer may be changed [22]. The increase in positive ions (holes) reduces the number of negative ions (electrons). Consequently, the n-Si layer, which functions as the donor, attracts more electrons and, therefore, functions as the acceptor. Constant voltage stress experiments on MOSCAPs have not been performed; according to the results, this is the most convenient and realistic explanation for this behavior. Therefore, there is a possibility of some changes in the lifetime of the MOS capacitors in terms of annealing [40]. The relation between lifetime and annealing conditions will be further discussed in an extension of this work.

Figure 6 presents the leakage current densities and breakdown voltage characteristic transformation in various PDA conditions after applying a positive bias voltage from 0 to 15 V. The sample 300 °C–5 min exhibits the lowest leakage current and highest breakdown voltage among all the samples; with increasing PDA time and temperature, the MOSCAP breaks down at a lower voltage. As a result, the sample 500 °C–5 min has the lowest breakdown voltage and highest leakage current. This behavior can be described with ion diffusion theory. Because the surface morphology has changed with increasing temperature, the interface between the oxide and semiconductor becomes leakier and cannot absorb the high bias voltage.

## 4. Conclusions

In this study, the characteristics of Al_2_O_3_ PDA annealed under different conditions on the Si substrate were investigated. The Al_2_O_3_ layers were deposited with H_2_O as an oxidant and ALD. The capacitance, hysteresis, dielectric constant, constant voltage stress, leakage current, breakdown voltage, interface trap, and border trap characteristics of annealed and nonannealed samples were compared. In addition, their surface morphologies were studied with XRD. The results clearly indicate that the annealed samples show better characteristics than the nonannealed samples. Moreover, the diffusion properties of the silicon and oxide layers were discussed. This article divulges an unopened scenario and also presents the optimal conditions for post-deposition annealing of Al/Al_2_O_3_/n-Si stacks.

## Figures and Tables

**Figure 1 materials-14-03328-f001:**
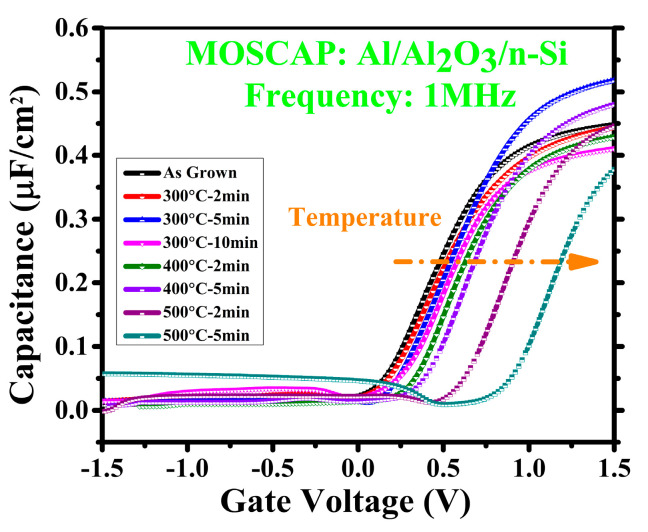
Capacitance–voltage curves comparison at 1 MHz.

**Figure 2 materials-14-03328-f002:**
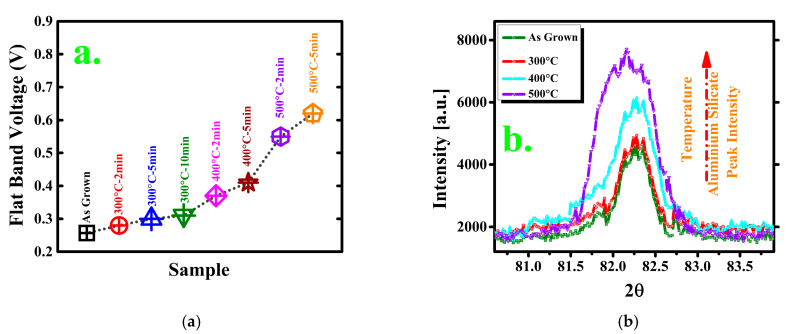
(**a**) Flat-band voltage comparison by inflection point method; (**b**) XRD peak value indicating aluminum silicate presence.

**Figure 3 materials-14-03328-f003:**
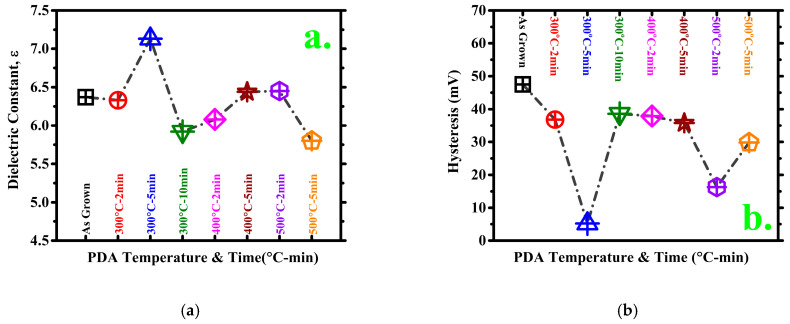
(**a**) Average dielectric constants; (**b**) comparison of average hysteresis.

**Figure 4 materials-14-03328-f004:**
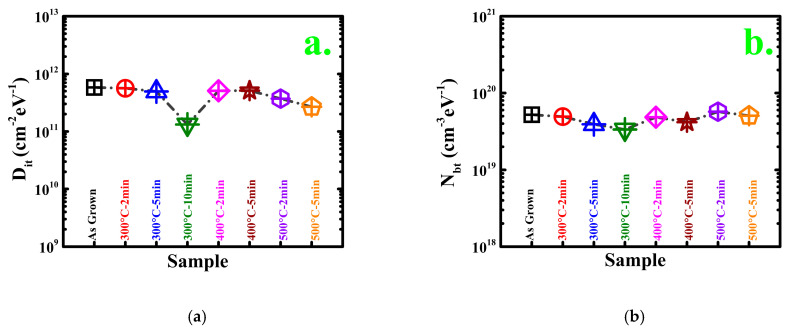
(**a**) Interface trap comparison and (**b**) border trap comparison for all samples.

**Figure 5 materials-14-03328-f005:**
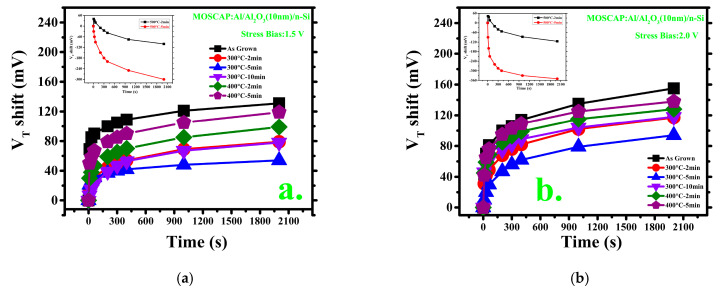
Threshold voltage shift after application of constant voltage stress for 2000s: (**a**) 1.5 V stress bias; (**b**) 2.0 V stress bias. Inset: 500 °C samples after stress application.

**Figure 6 materials-14-03328-f006:**
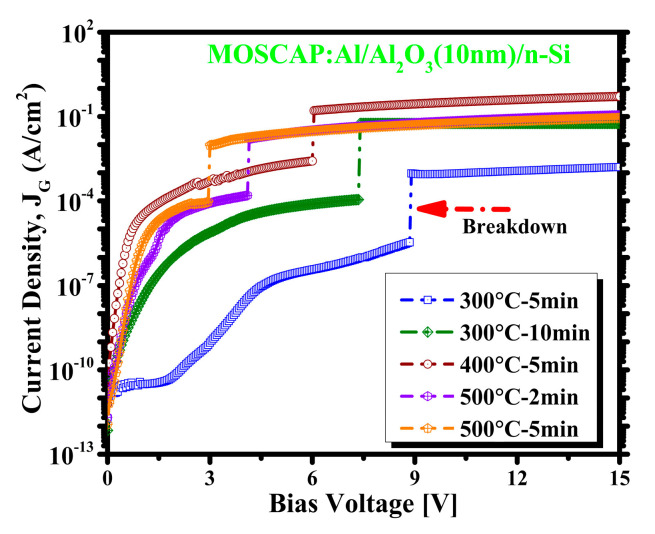
Breakdown voltage and leakage current of MOSCAPs.

**Table 1 materials-14-03328-t001:** Conditions of parameters used for MOSCAP fabrication.

	ALD			RTA			Thermal Evaporator
Sample	Oxide	Cycle	Temperature	Gas	Temperature	Time (min)	Metal Layer
As-grown	Al_2_O_3_	100	250	Ar	-	-	Al
300 °C–2 min		100	250	Ar	300 °C	2	
300 °C–5 min		100	250	Ar	300 °C	5	
300 °C–10 min		100	250	Ar	300 °C	10	
400 °C–2 min		100	250	Ar	400 °C	2	
400 °C–5 min		100	250	Ar	400 °C	5	
500 °C–2 min		100	250	Ar	500 °C	2	
500 °C–5 min		100	250	Ar	500 °C	5	

**Table 2 materials-14-03328-t002:** Highest capacitance values of tested devices.

	As-Grown	300 °C–2 min	300 °C–5 min	300 °C–10 min	400 °C–2 min	400 °C–5 min	500 °C–2 min	500 °C–5 min
Test 1	0.46	0.47	0.54	0.43	0.42	0.47	0.45	0.48
Test 2	0.38	0.52	0.53	0.45	0.45	0.47	0.53	0.52
Test 3	0.375	0.34	0.53	0.43	0.42	0.51	0.52	0.46
Test 4	0.48	0.46	0.53	0.38	0.51	0.50	0.43	0.29
Test 5	0.475	0.4	0.53	0.46	0.43	0.46	0.54	0.52
Test 6	0.375	0.35	0.56	0.40	0.44	0.44	0.48	0.27
Test 7	0.45	0.45	0.475	0.48	0.50	0.52	0.475	0.425
Avg	0.43	0.43	0.53	0.43	0.45	0.48	0.49	0.42
STD	0.05	0.07	0.03	0.03	0.04	0.03	0.04	0.1

**Table 3 materials-14-03328-t003:** Hysteresis data of tested devices.

	As-Grown	300 °C–2 min	300 °C–5 min	300 °C–10 min	400 °C–2 min	400 °C–5 min	500 °C–2 min	500 °C–5 min
Test 1	30.0	13.6	10.9	17.6	42.0	33.4	9.0	13.2
Test 2	55.5	31.3	2.2	40.7	56.6	58.2	14.6	22.2
Test 3	55.6	36.8	8.2	34.1	32.1	36.5	26.5	76.2
Test 4	53.7	6.1	2.4	54.3	28.1	28.7	2.5	9.4
Test 5	52.6	62.9	1.9	65.0	23.5	27.0	33.7	15.7
Test 6	51.1	36.8	0.4	33.5	49.0	44.5	3.9	10.4
Test 7	34.3	70.7	10.4	27.2	34.0	22.3	24.1	61.8
Avg	47.5	36.9	5.2	38.6	37.9	35.8	16.3	29.8
STD	10.7	23.6	4.5	16.1	11.8	12.2	12.0	27.4

## Data Availability

The data presented in this study are available on request from the corresponding author. The data are not publicly available due to privacy issues.

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
