# Peer review of "Extensive Analysis on the Effects of Post-Deposition Annealing for ALD-Deposited Al2O3 on an n-Type Silicon Substrate"

_materials, 2021, doi:10.3390/ma14123328_

Round 1
Reviewer 1 Report
The manuscript by A. Bhattacharjee and T.W. Kim presents ALD preparation of Al2O3 films on Si substrate and their electrical characteristics, including capacitance, dielectric constant and other. The text is well-written, however, major revisions should be performed to improve the presentation.
In title: "300-mm" is 30 cm, which is apparently a mistake.
In abstract: the MOSCAP abbreviation should be described.
In Materials&Methods section: "gas flow rates.. 300 sccm" - please, provide the conventional units [l/min]. "Growth per cycle was 1.14" - describe, what is GPC exactly? Is it dimensionless?
In Materials&Methods section: the description of XRD experiment is totally missed.
In Materials&Methods section: there is a table with no title, which is not mentioned in the text. Also in this table, the gas carrier is N2, however, in the text it is argon. Please, correct these discrepancies.
Figure 2b: there is no description of XRD experiment. Why do you show such big 2Theta around 80 deg? Which radiation was used in XRD? Which "Silicate" do you mean? The XRD results should be compared with database, and the proper description should be provided.
Tables 1 and 2: These tables provide the average values. The averages should be presented with their standard deviations, and the corresponding conclusions should be discussed in the text.
Author Response
Reviewer 1
Reviewer wrote:
The manuscript by A. Bhattacharjee and T.W. Kim presents ALD preparation of Al2O3 films on Si substrate and their electrical characteristics, including capacitance, dielectric constant and other. The text is well-written; however, major revisions should be performed to improve the presentation.
Our response:
Dear the reviewer,
Thank you very much for carefully reviewing our manuscript and providing fruitful suggestions. We have taken all the comments into consideration, as below. We hope that the revision would be satisfactory to the reviewer and looking forward to hearing more comments.
Corresponding change in manuscript: No
Comment 1
Reviewer wrote:
In title: "300-mm" is 30 cm, which is apparently a mistake.
Our response:
Thank you to the reviewer for this valuable suggestion to improve the quality of our manuscript. The title is revised and corrected.
Corresponding change in manuscript: Yes.
New: Title; “Extensive Analysis on the Effects of Post Deposition Annealing for ALD-deposited Al2O3 on an n-type Silicon Substrate.”
Location of change:
Section: Title
Comment 2
Reviewer wrote:
In abstract: the MOSCAP abbreviation should be described.
Our response:
The abbreviation has been included into the abstract of the revised manuscript.
Corresponding change in manuscript: Yes.
Location of change:
Section: Abstract
Page-1, line 17-18
Comment 3
Reviewer wrote:
In Materials & Methods section: "gas flow rates.. 300 sccm" - please, provide the conventional units [l/min]. "Growth per cycle was 1.14" - describe, what is GPC exactly? Is it dimensionless?
Our response:
As suggested by the reviewer, we have changed the gas flow rate unit from sccm (Standard Cubic Centimeter per Minute) to L/min (Liter per minute).
Growth per Cycle or GPC is defined as the incremental increase in the thickness of the film per cycle of deposition. The mathematical expression of GPC is like:
GPC= Film Thickness(measured by ellipsometer after deposition)/ Deposition Cycle(100 in our case)
It is denoted by Å (1Å = 10-10meter), which is added in our revised manuscript.
Corresponding change in manuscript: Yes.
New: Gas flow unit has been changed from sccm to L/min and dimension of GPC (Å) is added.
Location of change:
Section: Materials and Methods
Page-3, line 130 and line 141
Comment 4
Reviewer wrote:
In Materials & Methods section: the description of XRD experiment is totally missed.
Our response:
We would like to thank the reviewer for this attention to the details. We have included the XRD experiment description in revised manuscript.
Corresponding change in manuscript: Yes.
New: Line: X-Ray Diffraction (XRD) was measured by X-Ray Diffractometer (Rigaku ULTIMA 4) where, Cu Kα radiation (40kV, λ=1.54Å) was used for measuring.
Location of change:
Section: Materials and Methods
Page-4, line 155-157
Comment 5
Reviewer wrote:
In Materials & Methods section: there is a table with no title, which is not mentioned in the text. Also, in this table, the gas carrier is N2, however, in the text it is argon. Please, correct these discrepancies.
Our response:
Thanks for your comment. The table has been corrected.
Corresponding change in manuscript: Yes, we have added a title and description for Table 1. Also, gas carrier in the table is changed into Ar.
Location of change:
Section: Materials and Methods
Page-4, line 157-161
Table 1: Conditions of parameters used for MOSCAP fabrication.
Comment 6
Reviewer wrote:
In Figure 2b: there is no description of XRD experiment. Why do you show such big 2Theta around 80 deg? Which radiation was used in XRD? Which "Silicate" do you mean? The XRD results should be compared with database, and the proper description should be provided.
Our response:
We would like to thank the reviewer for this attention to the details. In XRD, we measured 2θ from 30Ëš to 90Ëš, but significant difference in the peak analysis, in terms of annealing, is not found except in 2θ = 82.5 Ëš. From the literature, we found, this value of 2θ, indicates the presence of aluminium silicate. That is why, we have narrated down the whole XRD waveform into around 80 degree.
Cu Kα radiation (40kV, λ=1.54Å) was used for measuring XRD.
By silicate, we meant Aluminium Silicate.
The XRD results are compared with references we provided and the description is included in our revised manuscript.
Corresponding change in manuscript: Suggested description with new references have been added.
New: We measured 2θ from 30Ëš to 90Ëš, but significant difference in the peak analysis, in terms of annealing, is not found except in 2θ=82.5 Ëš. From the literature [34-36], we found, this value of 2θ, indicates the presence of aluminium silicate. That is why, we have narrated down the whole XRD waveform into around 80 degree.
- Khalil, O.M.; Mingareev, I.; Bonhoff, T.; El-Sherif, A.F.; Richardson, M.C.; Harith, M.A. Studying the effect of zeolite inclusion in aluminum alloy on measurement of its surface hardness using laser-induced breakdown spectroscopy technique. Optical Engineering 2014, 53, 014106.
- Atuanya, C.U.; Ibhadode, A.O.A.; Dagwa, I.M. Effects of breadfruit seed hull ash on the microstructures and properties of Al-Si-Fe alloy/breadfruit seed hull ash particulate composites. Results in Physics 2012, 2, 142–149.
- Ma, P.; Jia, Y.; Prashanth, K.G.; Yu, Z.; Li, C.; Zhao, J.; Yang, S.; Huang, L. Effect of Si content on the microstructure and properties of Al-Si alloys fabricated using hot extrusion. Journal of Materials Research 2017, 32, 2210–2217.
Location of change:
Section: Results and Discussions
Page-5, line 184-188
Figure 2(b).
Section: References: 35 and 36
Page-11, line 381-384
Comment 7
Reviewer wrote:
Tables 1 and 2: These tables provide the average values. The averages should be presented with their standard deviations, and the corresponding conclusions should be discussed in the text.
Our response:
As suggested by the reviewer, standard deviation along with the corresponding conclusion has been added in revised manuscript.
Corresponding change in manuscript: Yes.
New: All test and average calculation results as well as the calculated standard deviation of the seven samples are listed in Tables 2 and 3. The standard deviation indicates how disperse the data is in relation to the average value. For example, in Table 2, the standard deviation values indicate that, 300°C_5min sample has the lowest scattering in terms of hysteresis.
Location of change:
Section: Results and Discussions
Page-6, line 206-209
Table 2 and Table 3.
Reviewer 2 Report
Bhattacharjee et al in the paper entitled "Extensive Analysis on the Effects of Post Deposition Annealing for ALD-deposited Al2O3 on a 300-mm Silicon Substrate" presented some important results
Suggest minor revision with the following comments.
Results and Discussion:
Why CV characteristics is at 1MHz? What about 1kHz or 10 kHz?
Please provide how VFB was calculated (say one line in the result section).
Graphs: Each graph has the same left Y-axis and Y-axis (Suggest removing the right one).
100 cycle ALD deposition was done. Are the properties changes with #cycle?
What about the lifetime? This paper discusses this. https://doi.org/10.1557/PROC-1153-A07-17 So, can annealing temperature change the lifetime?
Otherwise, the paper is OK.
Author Response
Reviewer 2
Reviewer wrote:
Bhattacharjee et al in the paper entitled "Extensive Analysis on the Effects of Post Deposition Annealing for ALD-deposited Al2O3 on a 300-mm Silicon Substrate" presented some important results
Suggest minor revision with the following comments. Otherwise, the paper is OK.
Our response:
Dear the reviewer,
Thank you very much for carefully reviewing our manuscript and providing fruitful suggestions. We have taken all the comments into consideration, as below. We hope that the revision would be satisfactory to the reviewer and looking forward to hearing more comments.
Corresponding change in manuscript: No
Comment 1
Reviewer wrote:
Why CV characteristics is at 1MHz? What about 1kHz or 10 kHz?
Our response:
In this work, we focused on 1MHz frequency, because in lower frequencies, CV curve usually get distorted by noise. Therefore, the actual characteristics are hard to get. In future, lower frequencies will also be discussed in the extension of this work.
Corresponding change in manuscript: No.
Comment 2
Reviewer wrote:
Please provide how VFB was calculated (say one line in the result section).
Our response:
Thank you for the comments. Flat Band voltage or VFB was calculated by inflection point method using a second derivative of the C-V curve. The point of intersection of second derivative of the CV curve with the Vg-axis is zero, which is called the inflection point as well as the value of flat band voltage.
Corresponding change in manuscript: Yes.
New: Flat Band voltage or VFB was calculated by inflection point method using a second derivative of the C-V curve. The point of intersection of second derivative of the CV curve with the Vg-axis is zero, which is called the inflection point as well as the value of flat band voltage.
Location of change:
Section: Results and Discussions
Page-4, line 177-179
Comment 3
Reviewer wrote:
Each graph has the same left Y-axis and Y-axis (Suggest removing the right one).
Our response:
Thanks for your valuable suggestion. We have revised our manuscript and have removed the right-side Y-axis. The purpose of giving same axis in both sides was for reviewers better understanding of the graph.
Corresponding change in manuscript: All Figures are changed accordingly.
Comment 4
Reviewer wrote:
100 cycle ALD deposition was done. Are the properties changes with cycle?
Our response:
There is no change in the properties in terms of cycle. However, since the cycle determines thin film thickness; if the thickness of oxide layer is very low, then there will be too much leakage current, band to band tunneling which means electron will pass through the insulator and move between metal and semiconductor. Therefore, in Atomic Layer Deposition, 100 cycle is considered as a standard value.
Corresponding change in manuscript: No.
Comment 5
Reviewer wrote:
What about the lifetime? This paper discusses this. https://doi.org/10.1557/PROC-1153-A07-17 So, can annealing temperature change the lifetime?
Our response:
Thanks for your valuable suggestion. From our constant voltage stress measurement results, we can say that, there is a possibility of some changes in lifetime of the MOS capacitors in terms of annealing. The relation between lifetime and annealing conditions will be further discussed in an extension of this work.
Corresponding change in manuscript: The above-mentioned information has added with the reference suggested by the reviewer.
New: Therefore, there is a possibility of some changes in lifetime of the MOS capacitors in terms of annealing [40]. The relation between lifetime and annealing conditions will be further discussed in an extension of this work.
- Wang, J.; Farrokh-Baroughi, M.; Shanmugam, M.; Samadzadeh-Tarighat, R.; Sivoththaman, S.; Paul, S. Passivation of silicon surfaces using atomic layer deposited metal oxides. Materials Research Society Symposium Proceedings 2009, 1153, 147–152.
Location of change:
Section: Results and Discussion
Page-8 and line 269-272.
Section: References: 40
Page-11 and line 390-392.
Round 2
Reviewer 1 Report
The manuscript has been properly revised by the authors. I recommend a publication in the present form.